# Real-Time Visualization of Cellulase Activity by Microorganisms on Surface

**DOI:** 10.3390/ijms21186593

**Published:** 2020-09-09

**Authors:** Pallavi Kumari, Tali Sayas, Patricia Bucki, Sigal Brown-Miyara, Maya Kleiman

**Affiliations:** 1Institute of Plant Sciences, Agricultural Research Organization (Volcani Center), Rishon Lezion 7505101, Israel; palcuj31@gmail.com (P.K.); talid@volcani.agri.gov.il (T.S.); 2Institute of Plant Protection, Agricultural Research Organization (Volcani Center), Rishon Lezion 7505101, Israel; pbucki@volcani.agri.gov.il (P.B.); sigalhor@volcani.agri.gov.il (S.B.-M.); 3Agro-NanoTechnology and Advanced Materials Center, Agricultural Research Organization (Volcani Center), Rishon Lezion 7505101, Israel

**Keywords:** hydrolysis, microorganisms, visualization, plant root, real-time

## Abstract

A variety of methods to detect cellulase secretion by microorganisms has been developed over the years, none of which enables the real-time visualization of cellulase activity on a surface. This visualization is critical to study the interaction between soil-borne cellulase-secreting microorganisms and the surface of plant roots and specifically, the effect of surface features on this interaction. Here, we modified the known carboxymethyl cellulase (CMC) hydrolysis visualization method to enable the real-time tracking of cellulase activity of microorganisms on a surface. A surface was formed using pure CMC with acridine orange dye incorporated in it. The dye disassociated from the film when hydrolysis occurred, forming a halo surrounding the point of hydrolysis. This enabled real-time visualization, since the common need for post hydrolysis dyeing was negated. Using root-knot nematode (RKN) as a model organism that penetrates plant roots, we showed that it was possible to follow microorganism cellulase secretion on the surface. Furthermore, the addition of natural additives was also shown to be an option and resulted in an increased RKN response. This method will be implemented in the future, investigating different microorganisms on a root surface microstructure replica, which can open a new avenue of research in the field of plant root–microorganism interactions.

## 1. Introduction

The first point of interaction between a microorganism and an associated plant occurs on the surface of the plant. The surface of the plant, while covered with waxes in some cases such as leaves of certain plant species, is composed mostly of cellulose [1,2,3]. Cellulose is the most abundant biopolymer on earth; it is one of the main structural components of plants, together with hemicellulose and lignin [4,5,6]. Cellulose is composed of D-glucose units covalently linked via β-1,4-glycosidic bonds [7,8], forming a spatial arrangement with crystalline and amorphous parts. The cellulose chains are arranged as a sheet within the crystalline fibers and connected with inter or intra molecular hydrogen bonds. Therefore, it is insoluble in water and many organic solvents [7]. During the interaction between the microorganism and plant surface, plant cellulose is often hydrolyzed. In particular, soil-borne plant pathogens often enter the plant through the cellulose-rich root surface, with the aid of secreted Plant Cell Wall Degrading Enzymes (PCWDE), including cellulase [9,10]. This phenomenon is of great interest to phytopathologists studying the complex set of interactions between microorganisms and plants. 

Cellulose hydrolysis can also be utilized for other purposes. For example, the enzymatic hydrolysis of cellulose into fermentable sugars and their subsequent conversion into a variety of potential chemicals and biofuels has been widely investigated for a long time [11]. The search for microorganisms to be used in cellulose hydrolysis for fuel production has yielded a variety of detection methods for cellulase activity [12]. Visualization of the hydrolysis process is a simple approach to achieve fast and semi-quantitative estimates of the hydrolysis. Surface-based cellulose hydrolysis detection methods have been developed over the years; however, using cellulose as a substrate is challenging, mostly due to its crystalline sites causing aggregation [7]. Hence, many of the detection methods largely rely on the visualization of hydrolysis of the cellulose derivative carboxymethyl cellulose (CMC) [13]. Indeed, bacteria with cellulytic activity isolated from plants were tested on CMC platforms to measure their cellulytic potential in various conditions that are similar and divergent from natural conditions [14,15,16].

CMC is the most widely used cellulose derivative for studying cellulytic activity by microorganisms [13]. In addition, it is widely used as a reagent for cellulase activity investigation in various other applications [16,17,18,19,20,21]. CMC is soluble in water, and this property is used to incorporate CMC into gel-like substrates such as agar or gelrite, which is the most common method for studying cellulase activity when using this derivative [22,23,24,25]. The microorganisms to be inspected are cultured on a gel-like medium containing CMC, leaving sufficient time for hydrolysis activity to occur. Then, the medium is dyed with a dye that interacts only with either CMC or the hydrolysis products, differentiating the hydrolysis zone by a changed color, enabling the visualization of the hydrolysis that occurred [26,27]. In the past, a variety of dyes have been introduced for post hydrolysis plate staining, the most common of which are Gram’s iodine and Congo red [27,28]. The use of these dyes poses several problems; for example, the films can be washed off after staining or the hydrolysis zone is not always easily discernible [29]. These issues restrict the ability to accurately quantify the hydrolysis. 

While the use of CMC and dye as a platform for cellulase detection in microorganisms has been investigated over the years for different applications, almost no attention was given to the spatial resolution of the microorganisms cellulase activity on plant tissues. Johnsen et al. discussed the importance of spatial resolution in the context of cellulase activity on plant tissue prints [28]. In this study, solidified CMC films and CMC in solution were used to better quantify the hydrolysis [28]. However, the dyeing process was still performed after the hydrolysis reaction was completed, eliminating real-time visualization of the hydrolysis process, which is a visualization that may teach us important lessons about the microorganisms’ spatial mode of operation. 

Spatial resolution can be crucial in the context of plant interactions with microorganisms and highly relevant for studying plant–pathogen interactions. For example, although some soil-borne pathogens are known to enter the root in specific locations [30,31,32], the structural basis for this preference has received little attention. Additionally, synthetic systems were proven efficient in depicting the structural preference in leaves [33,34]. We recently introduced a synthetic method to study structural preferences in soil-borne pathogens [35]. This method involves the replication of the root surface microstructure in a synthetic system in order to study the spatial preferences in root pathogens localization on the root surface. We demonstrated the replication of tomato root surface microstructure using the silicone-based polymer polydimethyl siloxane (PDMS). However, PDMS replicated root surfaces do not interact with the pathogen and are mostly valuable in determining pathogen localization. Since this method is not limited in the choice of material and any material that can generate a film can be used to replicate the root surface microstructure, cellulose-based materials are the next logical step.

Here, we develop a novel method for cellulase activity visualization. This method is based on CMC films tagged by acridine orange, a biocompatible dye, for the real-time screening of cellulase activity. This method is a cheap and easy way to study cellulytic activity in real time. In this study, we tagged CMC with acridine orange through an adsorption process, generating dyed films. We chose CMC as our surface material, since it forms a 3D gel-like structure in an aqueous environment and hence can be used in its pure form to investigate cellulase activity [36], specifically with a spatial resolution that might be prevented when mixed with other materials [28]. We chose acridine orange, a versatile fluorescent dye that is mostly used for cell staining as well as for RNA and DNA staining in living cells and gels [37], as a dye to visualize hydrolysis, since it is a cationic dye and therefore is highly soluble in water and compatible with CMC [38]. Additionally, it enables the incorporation of microorganisms into the system, since its toxicity is limited [39]. The tagging of CMC with the dye prevents the need for post hydrolysis staining, enabling a reliable and reproducible visualization and detection under real-time conditions of enzymatic hydrolysis. Both synthetic cellulase and a live root pathogen representative (root-knot nematode—RKN) were used under various parameters. This method has the potential to be expanded to detect the spatial distribution of cellulytic activity of many living microorganism sources. For example, a dyed CMC based root surface replica could be used both as a structural replica of the root and as a reactive surface, upon which cellulose hydrolysis by cellulase secreted from soilborne pathogens can be investigated.

## 2. Results

### 2.1. Enzyme Hydrolysis of Tagged CMC Films

Our goal was to establish a system that enables the real-time visualization of cellulase secretion by microorganisms on surfaces, allowing to later proceed to biomimetic plant surfaces. The visualization of enzymatic hydrolysis of CMC films by synthetic cellulase is shown in Figure 1, under hydrolysis optimal parameters [40]: pH value of 5 and incubation at 37 °C for 2 h. This hydrolysis produced a clear halo where the enzyme was added, which was not visible in the control film (Figure 1). The halo formation was due to CMC hydrolysis, resulting in the release of incorporated dye. Two other dyes examined were found to be inferior to acridine orange. Both are widely used in the literature and are often compared [27,29,41]. Congo red was easily washed out of the film, regardless of cellulase activity and using Gram’s iodine resulted in an unclear hydrolysis zone.

To establish that the formed halo is indeed the result of CMC hydrolysis, Benedict’s test [42] was performed to detect the presence of reducing sugars (such as glucose). The reaction between the sugar and the sodium ion in Benedict’s reagent causes the formation of enols. In turn, Cu^2+^ present in the test solution is reduced to Cu^+^, resulting in a color change and a precipitation occurring due to insolubility in water upon boiling [42]. The color varies from green to dark red or rusty-brown, depending on the amount and type of sugar. For the purpose of Benedict’s test, we allowed the reaction to continue for 18 h, to generate a sufficient amount of glucose for detection. Both the control and the reacted film were washed with a measured amount of water. Initially, the water in the enzyme treatment (solution II) was slightly colored as opposed to water from the control film (solution I) due to the release of acridine orange dye from the film (Figure 1A). Then, the wash solutions were subjected to Benedict’s test, and a distinct rusty red precipitate appeared in solution II, indicating the presence of reducing sugars in the water that originated from the reaction of the film, but not in the control wash (Figure 1B). A bright blue color was observed in the control wash due to the presence of Cu^2+^ in Benedict’s solution. 

### 2.2. Quantification of Products as a Function of Enzyme Concentration

To evaluate the range of linearity of the hydrolytic activity visualization, we performed a dose–response assay with increasing enzyme concentrations. Enzyme concentrations ranging from 1 to 30 μg/10 μL were used, keeping the volume constant at 10 μL, eliminating possible liquid volume effects. The films were prewashed with acetate buffer at pH 5 and incubated at 37 °C for 2 h. Representative images showing the halo size and color resulting from various enzyme concentrations are shown in Figure 2A. As can be seen from the images, both size and hues change with increasing amounts of enzyme. All films started out uniformly at a thickness of 1 mm. Film hydrolysis leads to a change in hue in the hydrolyzed region, which is probably due to the consumption of film thickness, and it reaches saturation when all the film thickness is consumed. Since saturation in hue change occurred before maximal enzyme amount, we chose the halo surface area for a reliable quantification. Surface area quantification represents the hydrolysis level, which in turn represents the amount of enzyme. The quantification of halo surface area as a function of enzyme concentration is shown in Figure 2B. The four higher concentrations, in which the hue was dominated by a white color, were chosen. The results show a linear correlation (R^2^ = 0.9745) between enzyme concentration and halo area. Since halo hue could not be used for the quantification of hydrolysis due to the saturation effect, we used UV-Vis spectroscopy of wash solutions for that purpose. As the hydrolysis of CMC results in the release of acridine orange from the film, the known maximum absorbance wavelength of acridine orange at 480 nm was used to assess the amount of dye released from the film. Then, we calculated what percentage of the acridine orange added to the film was released during the hydrolysis (Figure 2C). This analysis shows that dye release increased with enzyme quantity (Figure 2C). The highest enzyme concentration (20 μg/10 μL) resulted in aggregated dye; this was probably due to the high concentration of dye released, which was probably incorporated in CMC polymeric chain and hence could not be diluted. Hence, the absorbance was lower than that detected with the lower concentration (15 μg/10 μL), and so 15 μg/10 μL was considered as the limit of detection using the acridine orange absorbance method.

To quantify in an additional mean, specifically in the concentration range that was not accessible to quantification using acridine orange absorbance, we analyzed the wash products using Sumner’s method. In this method, 3,5-dinitrosalicylic acid is subjected to the tested solution. In the presence of reducing sugars, and upon boiling, 3,5-dinitrosalicylic acid is reduced to 3-amino-5-nitrosalicylic acid, which strongly absorbs light at 540 nm. To perform this assay, undyed CMC films were used, to prevent the noise in absorbance that may occur from the dye. We also performed the reaction for 16 h, since enough reducing sugars needed to be produced to overcome the sensitivity of the assay. We calculated the weight of reducing sugars released as a percentage of the CMC in the original film (Figure 2D). This calculation showed that at low enzyme concentrations, the amount of products was undetectable using this method, but as the enzyme concentration went up, so did the amount of reducing sugars (Figure 2D). At the standard concentration of 10 μg/10 μL, we saw that roughly 4% of the CMC in the plate was converted to glucose. When using the known amount of CMC added to the plate, this means that 20 units of enzyme were applied, which correlates with the manufacture report regarding specific enzyme activity. All measurements gave roughly the same percentage of CMC hydrolysis, including the halo, when calculated as percentage of total area (data not shown). Quantification of the enzymatic reaction is often performed to assess microorganisms’ hydrolysis ability. The quantification is usually performed by measuring the hydrolysis products and is mostly done in solution using multiple techniques [40,43,44,45]. Some methods were developed for hydrolysis quantification on a surface by measuring the halo size, and indeed, a correlation was found between enzyme units and halo size. However, this was performed on an agar plate mixed with CMC, which presents a different reaction and diffusion pattern than the ones developed here [46,47]. We showed that activity is increasing with increasing enzyme concentration, using various methods, but more importantly, we showed that the most sensitive and accurate method is the visualization method, which is the method developed in this assay. The halo was consistently larger with increasing enzyme concentration and was clearly visible and quantifiable at both low and high enzyme concentrations. The aim of this method is not to use the halo size as quantification between different assays, as it may vary depending on the conditions of the assay. However, we have shown that it can be used to compare two different experiments performed at the same conditions.

### 2.3. Change in Halo Size over Time

In order to observe the real-time visualization of CMC hydrolysis, halo development was followed over time. Figure 3 shows the enzymatic hydrolysis of the film at different time points, between 5 min and 2 h. A clear halo appeared in the film as early as 5 min after cellulase addition (Figure 3). While the halo size mostly changed in the first hour and remained stable afterwards, a gradual change in film color was observed over time until a complete loss of orange color is achieved after 2 h. This loss of color converts into a hole in the film with additional incubation time. The goal of this experiment was to show the feasibility of this system in the real-time visualization of cellulase secretion for future studies, examining the cellulase secretion of live microorganisms. The detection of cellulase activity in other studies has been performed in various methods, some of which occur in a liquid environment, and some of which occur on a plate surface. Either way, the detection for cellulase activity requires additional steps such as viscosity measurements [48,49], detection of hydrolysis products [43,45], adding dye to plates [27,29,50] and others [21,40,51]. Incubation time for the hydrolysis reaction can vary between minutes and days, but the detection is not performed simultaneously with the reaction. This is because the purpose of those studies is to find better, faster, and more accurate methods to detect cellulase activity in microorganisms. However, our goal is to track this activity to better understand the effect of the surface on the microorganisms activity. Hence, for this purpose, a real-time visualization is critical. 

### 2.4. Hydrolysis Reaction under Different Temperature and pH Conditions

To further characterize the system, we tested the reaction at different temperatures and pH values. Initial studies were performed at pH = 5 and 37 °C, as these are the ideal parameters for the specific cellulase variety we used, according to manufacturer. Ideal pH and temperature conditions for the hydrolysis reaction highly depend on the source of cellulase. Indeed, many studies performed on different cellulase sources showed different ideal temperature and pH conditions [17,19,20,21]. While mostly ideal pH conditions were around pH = 5, temperature conditions varied greatly between 30 and 50 °C. It was important to us to show that this system can be sensitive enough to show hydrolysis in conditions relevant to the tested microorganisms. Temperature was varied by 10 °C in each direction (Figure 4A). 27 °C is an important temperature to examine, since this is the ideal temperature for the model organism tested (RKN) [52]. We found, as expected, that at this temperature, the hydrolysis is less efficient. Nonetheless, halo is clearly visible and hence quantifiable, although lowering the temperature may affect the sensitivity of the assay (Figure 4A). Since soil can reach very high temperatures during the summer, which can affect microorganism behavior [53], we also tested hydrolysis visualization at 47 °C. This temperature was also less efficient than 37 °C but to a lower extent than 27 °C (Figure 4A). This demonstrates that the newly developed assay can be used in a wide range of temperatures according to the optimal environmental temperature of the microorganism tested. We next tested CMC hydrolysis at various pH, ranging between 3 and 6 (Figure 4B). The reaction occurred at all pH values tested, although it was less efficient when deviating from the optimal pH. Changing the conditions to more acidic conditions decreased the efficiency more than changing to more basic conditions (Figure 4B). Since biological microorganisms have a wide range of pH preferences, this means that the newly established method can be used to test cellulase activity in a wide range of microorganisms. The qualitative phenomenon observed using the stereomicroscope was also quantified using UV-vis measuring the amount of acridine orange released (Figure 4C). The quantitative data match the qualitative data.

### 2.5. CMC Film Hydrolysis by Root-Knot Nematode (RKN)

To test the newly developed visualization technique in a physiologically relevant scenario, we chose to visualize interactions with root-knot nematode (RKN). RKN is a widespread root pathogen that penetrates the root by secreting Plant Cell Wall Degrading Enzymes (PCWDE) including cellulase [54,55,56], causing substantial damage to crops around the world [30,57,58,59,60]. We applied RKN to the center of the dyed CMC film and incubated for 2 h at either 37 °C (ideal temperature for the hydrolysis reaction) or 27 °C (ideal temperature for RKN). The color change pattern differed from the one observed with the application of pure enzyme. Unlike the pure enzyme, there was no single halo formed in the film; rather, there were many small halos correlating to nematode location on the film (Figure 5A). This shows that diffusion of the reaction is well inhibited, making the spatial sensitivity of the assay sufficient to detect a single nematode. Indeed, the number of halos correlates with the number of RKNs applied (Figure 5A). The halos at 27 °C were smaller than the ones formed at 37 °C. We believe this is the result of lower efficiency of the hydrolysis reaction at 27 °C. It should be noted that RKN is an obligate parasite, meaning it cannot survive for a long time outside the plant [61]. Under ideal in vitro conditions, kept at a low temperature, RKN will survive for a few days. Under the conditions used here, at a higher temperature and, more importantly, at relatively dry conditions, RKN cannot survive for more than a few hours. Hence, we did not expect a high motility of RKN. However, we observed that RKN actively secrete cellulase for at least 2 h, even at the higher temperature. Since a lower temperature significantly reduced the efficiency of the reaction, this could damage the resolution and ability to view the reaction at early stages and should be addressed later on, probably either by applying less than ideal conditions to the RKN, in which their functionality was already established, or by using higher resolution microscopy methods. During the development of this method, we only used stereomicroscope imaging, as we aimed at keeping the system’s simplicity. However, a higher resolution microscope will allow to better view the reaction at lower efficiencies.

To further explore cellulase secretion activity by microorganisms under natural-like conditions, specifically the influence of added natural additives, we added root extract as a biochemical factor, simulating root-secreted factors. Dyed CMC films with added root extract was subjected to nematodes in parallel to films with no root extract added to it (Figure 5B). Root extracts of target plants are known to influence RKN behavior and pathogenesis and are widely investigated [62,63,64]. Additional work is being performed on natural additives as a means of RKN control [59,65,66]. In this system, the addition of root extract to the film resulted in a higher CMC hydrolysis, and it was visualized by a clear halo on the film, similar to the halo formed by using a purified enzyme. This contrasted with RKN behavior in the absence of root extract, where several small halos were formed in areas of cellulase secretion (Figure 5B). We assume that the halo was formed due to a high secretion of cellulase, resulting in the formation of a distinct, singular large halo. This can be evident by the uneven color of the halo, which is lighter where the nematodes—the sole source of cellulase in this assay—are located. Additionally, it is possible that the addition of root extract extended the viability of the nematodes due to more natural conditions. When subjecting the CMC film with root extract to pure enzyme, no difference in reaction, compared to film with no extract added, was observed (data not shown). This supports the claim that the difference observed in Figure 5B originated from the nematodes and not from a change in the film hydrolysis reaction due to the addition of root extract. These results demonstrate the ability to visualize different degrees of cellulase activity by microorganisms and to incorporate relevant natural additives into the system to better depict the specific contribution of different factors. 

### 2.6. Surface Characterization of Hydrolyzed and Non-Hydrolyzed Films

Different hydrolysis conditions can affect surface structure; so, we analyzed the surface of the films after they were subjected to enzymatic hydrolysis. We used SEM to view a non-hydrolyzed film and compared it to the surface of films subjected to 10 µg of cellulase or RKN (Figure 6). The surface of unreacted film was completely smooth (Figure 6A). In contrast, both the film subjected to cellulase (Figure 6B) and the film subjected to RKN (Figure 6C,D) show rough areas. The pattern of the areas was different between the two treatments (compare Figure 6B to Figure 6C,D). In the RKN treatment, the rough areas probably represent nematode location and are similar in size to the ones detected under the stereomicroscope (compare Figure 5A to Figure 6C). However, the roughness formed by the cellulase enzyme is more dispersed upon the surface (Figure 6B). SEM visualization of various CMC films was performed in the past, especially in the context of edible and anti-microbial film, which are incorporated with nanoparticles or essential oils for that purpose [67,68,69,70]. Hence, SEM images of pure CMC films are scarce, yet the few found resemble the films we observe, and slight changes can be attributed to the acridine orange added to the films here.

To quantify the qualitative phenomenon observed with the SEM, we further studied the different surfaces using atomic force microscopy (AFM) analysis, which is ideal for the quantitative measurement of surface roughness and nanotexture. We performed AFM on a non-hydrolyzed CMC film as well as films subjected to both 10 μg of enzyme and RKN (Figure 7). Using AFM, it was shown that the untreated CMC film had a regular pattern with small periodic holes about 10 nm in size (Figure 7A). AFM measurements on pure CMC films have been performed in the past, especially in the context of nanoparticles incorporation [71,72]. However, it is important to mention that the roughness of the surface will highly depend on the particular setting, specifically the viscosity and modification ratio of CMC used as well as the percentage of CMC dissolved in the solvent. In conditions similar to the ones used here, the same periodic pattern was observed in pure CMC film [71], although we could not find any data regarding films hydrolyzed by cellulase. We calculated the roughness of the surface and found that the Root Mean Square (RMS) of the pure CMC film is 17.9 nm (Figure 7A). When the film was subjected to hydrolysis, from any of the two sources used (enzyme or RKN), the roughness increased (Figure 7B,C). Films subjected to cellulase showed a variety of hole sizes, with no specific pattern (Figure 7B, marked with green arrows). The holes in the film subjected to RKN were smaller than the ones in the film subjected to cellulase, but they were still larger than those in the untreated film (Figure 7C, marked with green arrows). Both films subjected to hydrolysis show a higher roughness than the untreated film. The RMS of the film subjected to nematodes was about twice that of the untreated film (34.4nm, Figure 7C). The RMS of the film subjected to enzyme had an even higher RMS (40.6 nm, Figure 7B). The same pattern was observed with the arithmetic average (Ra). This may suggest that the hydrolysis by RKN is more spread out on the surface with a lower penetration into the film as compared to the pure enzyme hydrolysis.

## 3. Discussion

In summary, we developed and analyzed a reliable and robust method for real-time visualization and the semi-quantification of cellulase activity by microorganisms. We used acridine orange with CMC to generate a film that changes color upon CMC hydrolysis. We characterized both the hydrolysis reaction and the surface of the films upon the reaction. We showed that it is possible to stray from ideal hydrolysis reaction conditions for microorganism-oriented conditions, in both temperature and pH. Additionally, we quantified the visualization of the hydrolysis as it correlates with enzyme concentration applied on the surface. We confirmed this by the quantification of reduced sugars generated from this hydrolysis. Furthermore, we characterized the topography of the surface before and after hydrolysis both qualitatively and quantitatively. Lastly, we used RKN as a model microorganism and showed that they are active on the film and can hydrolyze it in a visible manner. We also showed that their activity is influenced by natural additives added to the film.

This method allows for a simple and fast detection of cellulase activity on films. This method also does not require post hydrolysis dyeing as most other methods [27,28,29,41], since the dye is already incorporated in the film. This is an improvement over previous methods, as it allows for the visualization of CMC hydrolysis in real time. This is important for our goal to develop this method to follow microorganism activity, i.e., cellulase secretion, as it occurs. 

This synthetic system is successful in testing the effect of different parameters on microorganism behavior, which we showed by the addition of root extract into the films and the response of RKN to this additive. Next steps will include the separation of root materials and the controlled addition of isolated substances. Additionally, work on adding those substances with a spatial resolution to further link between topographical (physical) and chemical effects on microorganismal behavior on surfaces will be performed.

Additional next steps would be to use actual cellulose instead of cellulose derivatives as film material. In that context, it is interesting to mention that the process of cellulose hydrolysis has been studied using AFM [73,74], showing the directionality of hydrolysis in the contexts of cellulose fibers orientation. Of course, this is different from what we observed in the CMC films (Figure 7), but it is critical once materials that resemble even more to the natural surface of roots are used.

This work is a continuation of our previous work [35], where a synthetic replication of the root surface microstructure was built to better understand how microtopography influences microorganism location on the root surface. This system was built as a simplified model of a root–microorganism interaction, focusing on the structural/physical aspect of this interaction. A critical part of the root–microorganism interaction is the penetration of the microorganism into the root, which is often performed by the secretion of cellulase and other enzymes [56,75,76,77]. A combination of the method developed here for a real-time visualization of cellulase secretion by microorganisms with the previous method we developed for root surface microstructure replication will lead to a new avenue in the study of root–microorganism interactions. The combination of the two methods can enable the real-time visualization of both microorganism location (through known tagging methods, if needed) and activity in the form of cellulase secretion on the root surface topography. This elaborated synthetic system can help researchers separate different parameters of this interaction, starting from the structural parameter and combining it with different chemical parameters, to get a better insight into the complex interactions between root and microorganisms occurring on the surface of the root. Such research can assist in finding new routes for pathogen resistance in plants. Structural-based resistance has not yet been studied and could lead to new avenues in the areas of pest control and specifically, biological pest control.

## 4. Materials and Methods

### 4.1. Preparation of Acridine Orange-CMC Films

First, 20 mg of acridine orange hemi (Zinc chloride) salt (A6014, Sigma Aldrich, St. Louis, MO, USA) dye was placed in 50 mL of distilled water and stirred for 60 min. In parallel, 2 g of carboxymethyl cellulose (CMC) sodium salt, medium viscosity (C4888, Sigma Aldrich, St. Louis, MO, USA) powder was thoroughly dispersed in 50 mL of distilled water and stirred for 60 min. After complete dispersion, the 50 mL of the CMC solution were added to the 50 mL of the dye solution. The whole mixture was stirred for 24 h at room temperature under dark conditions. Then, 10 mL of the CMC sodium salt tagged with dye solution was spread onto a petri dish. The solution was dried under a hood overnight, and a smooth film was obtained. All further investigations were performed on the films within the petri dishes.

### 4.2. Buffer and Enzyme Preparation

First, 0.1 M of acetate buffer was prepared by fully dissolving 4 g of sodium acetate anhydrous (GR, Sinopharm Chemical Reagent Co., Ltd., ShangHai, China) in 400 mL of deionized water. Then, 0.9 mL of acetic acid (AR, Beijing Tongguang Fine Chemicals Co., Ltd., Beijing, China) was added to the solution. Then, the solution was made up to a total of 500 mL using deionized water. The pH of the resulting buffer solution was approximately 5. Enzyme solution was prepared by dissolving the enzyme (Cellulase onozuka from T. reesei C8546-10KU, ATCC2692, 0.47 FPU/mg, Sigma-Aldrich) in the buffer solution to all desired concentrations used in the experiments. According to the manufacturer, the enzyme has a specific activity of 2 units/1 µg (unit being the release of 1 µmol of glucose from cellulose in 1 h at pH 5 and 37 °C in a 2 h incubation). Hence, in the standard assay, where a 10 μL solution in a 1 µg/1 μL concentration was used, this equals 20 units.

### 4.3. CMC Film Hydrolysis

First, 20 mL of acetate buffer (0.1M) was poured on two films. After 1 min, the films gained a gel-like texture, and the buffer solution was removed and replaced by 10 µL of either buffer solution or enzyme solution (1 µg/1 µL), which were added onto the center of the film. Both films were incubated at 37 °C for 2 h. After 2 h, the films were washed with 5 or 20 mL water each and dried under the hood for 2 h. The washing solutions were kept in a vial for further analysis. Then, the films were visualized using a Nikon smz 25 stereomicroscope (Nikon Instruments Inc. NY, USA) equipped with NIS-elements software, and an image was captured using Nikon DS-Ri2 camera.

The same method was used to assess the effect of different parameters on the film hydrolysis:Time: Six CMC films were incubated at 37 °C and removed at different time points: after 5 min, 15 min, 30 min, 60 min, 90 min, and 120 min.Enzyme concentration: Seven different concentrations of cellulase enzyme were prepared in acetate buffer (1, 2, 5, 10, 15, 20, and 30 µg in 10 μL). Solutions with different enzyme concentrations were added to four different locations on the same film. The film was incubated at 37 °C for 2 h.Temperature: Three CMC films were used for the temperature-dependence experiment. The experiment was performed with 10 µg of enzyme and an incubation time of 2 h. The films were incubated at 3 different temperatures: 27 °C, 37 °C and 47 °C.pH: Four acetate buffer solutions at different pH values (3, 4, 5, 6) were prepared by titration with HCl or KOH. Four separate dyed CMC films were taken, and each one was washed with one of the buffers for 1 min. The experiment was performed with 10 µg of enzyme and incubation for 2 h at 37 °C.

All experiments were run in triplicate. 

### 4.4. Benedict’s Test

Reaction products in film wash (20 mL) of either buffer or enzyme were concentrated to 2 mL by evaporation at 60 °C. Then, the solution was cooled down to room temperature. Then, 0.2 mL of Benedict’s reagent (Sigma Aldrich, Dorset, England) was added to the vial. The presence of the alkaline sodium carbonate in the reagent converted the reducing sugars into enediols, which in turn decrease the cupric particles (Cu^2+^) present in Benedict’s reagent to cuprous particles (Cu^+^); upon heating, these appear as red copper oxide (Cu_2_O), which is insoluble in water. The vial was then boiled for 3–4 min. Then, color change was qualitatively examined.

### 4.5. Sumner’s Method

Sumner reagent was prepared by adding 500 mL of deionized water to a 2 L Erlenmeyer flask and heating on a hot plate at a low temperature (about 40 °C). Then, 300 g of potassium sodium tartarate (S-2377 Sigma Aldrich, St. Louis, MO, USA) was added and mixed. Then, 200 mL of NaOH 2N (MERCK 6498) was added. Next, 10 g of 3,5-dinitrosalicilic acid (D-0550 Sigma Aldrich, St. Louis, MO, USA) was added and dissolved completely. The solution was cooled and completed to 1 L with dH_2_O. Then, the solution was kept in a dark bottle at room temperature for later use. To quantitatively test the reducing sugars in the wash solution, 0.5 mL of reaction products in film wash (5 mL) of different enzyme concentrations were added to 0.5 mL of Sumner reagent. The mixture was incubated in a boiling water bath for 5 min. Then, color change was examined using UV-vis absorbance at 510 nm. Absorption studies were carried out at room temperature on a UV-vis spectrophotometer (Hewlett-Packard, model 8453, Agilent Technologies, CA, USA) from 200 to 600 nm using a quartz cell (10 mm path length). For calibration, known amounts of glucose were subjected to the same process. Then, the amount of glucose in the wash solution was calculated and given as the percentage of the original CMC weight added to the film.

### 4.6. Acridine Orange Measurements

Known amounts of acridine orange were dissolved in 2 mL of deionized water. Quantification was performed at 480 nm, the λ max of acridine orange. The measured absorbance was used to generate a calibration curve, which was based on Beer’s law. To measure the amount of acridine orange released during film hydrolysis, CMC films were washed with 5 mL of water, which were stored at room temperature for later use. The film washes of different concentrations were analyzed by absorbance measurement using UV-vis spectroscopy. Background UV-vis spectra of the water used for initial film wash were subtracted from the spectra of the cellulose hydrolysate solution. The dye concentration in each solution was calculated based on the calibration curve and given as a percentage of dye added to the film.

### 4.7. Root-Knot Nematode (RKN; Meloidogyne Javanica) Preparation

RKN were propagated on greenhouse-grown tomato ‘Avigail’ (870) plants. Nematode egg masses were extracted from roots with 0.05% (*v*/*v*) sodium hypochlorite followed by sucrose flotation. For sterilization, eggs were placed on a cellulose acetate filter membrane (Sartorius Stedim Biotech GmbH, Goettingen, Germany, pore size 5μm) in a sterile Whatman^®^ filter holder (Whatman International Ltd., Dassel, Germany). Eggs on the filter were exposed for 10 min to 0.01% (*w*/*v*) mercuric chloride (Sigma-Aldrich, St Louis, MO, USA), followed by 0.7% (*v*/*v*) streptomycin solution (Sigma-Aldrich), and three washing steps with 50 mL of sterilized distilled water. The sterilized eggs were collected from the membrane and placed on 25-μm pore sieves in 0.01 M 2-morpholinoethanesulfonic acid buffer (Sigma-Aldrich) under aseptic dark conditions for 3 days, allowing J2s to hatch. For the RKN, J2 is the infective stage that penetrates the root elongation zone toward the vascular cylinder to initiate infection. Freshly hatched preparasitic J2s were collected in a 50 mL falcon tube. For the visualization of the RKN cellulase secretion test, 50 μL containing approximately 80 nematodes were added to the center of dyed CMC films pre-treated by acetate buffer as described above. Films were incubated at either 27 °C or 37 °C for 2 h and then thoroughly washed with water 2–3 times. Then, the films were used for visualization by stereomicroscope, scanning electron microscopy (SEM), and atomic force microscopy (AFM). 

### 4.8. Root Extract Addition to CMC Film

Tomato cultivar M82 roots were physically crushed using a mortar and pestle. The resulting solution was filtered using gauze. The filtered liquid was used instead of water in the dissolution process of CMC during the preparation of the dyed CMC film. Then, the film was prepared and exposed to RKN as described above.

### 4.9. Scanning Electron Microscopy (SEM) Visualization

Scanning was performed by S-3400N, Hitachi, Tokyo, Japan at an accelerating voltage of 20 kV. The surface was coated by gold using a sputter coater (E-1010, Hitachi, Tokyo, Japan) before visualization.

### 4.10. Atomic Force Microscopy (AFM) Visualization

AFM topographic imaging was performed using an Innova Atomic Force Microscope with a NanoDrive Controller (Bruker, CA, USA). Small pieces of film were glued onto metal disks and attached to a magnetic sample holder located on the top of the scanner tube. Phase images were recorded under ambient air conditions. AFM images were taken with a scan rate of 1.00 Hz using tapping mode. Scan sizes were 1–10 µm with 256 scans on each scanning line. 

## Figures and Tables

**Figure 1 ijms-21-06593-f001:**
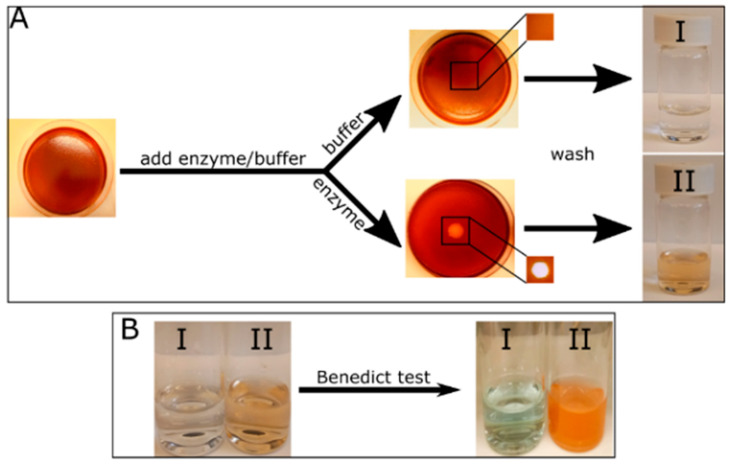
Enzymatic hydrolysis of carboxymethyl cellulase (CMC) film tagged with acridine orange. Dyed CMC films, pre-treated with acetate buffer, were subjected to either 10 μL acetate buffer (I) or 10 μL containing 10 μg cellulase (II). The films were incubated at 37 °C for 2 h, after which a clear halo was formed only on the film subjected to the enzyme (**A**, middle section, with enlargement of relevant area). Then, the films were washed with 20 mL of distilled water, which was collected into a vial. Only the water collected from the film subjected to the enzyme showed a change in color (**A**, right side). Then, the washed water was subjected to Benedict’s test, clearly showing the presence of sugars only in the washed water from the film subjected to the enzyme (**B**).

**Figure 2 ijms-21-06593-f002:**
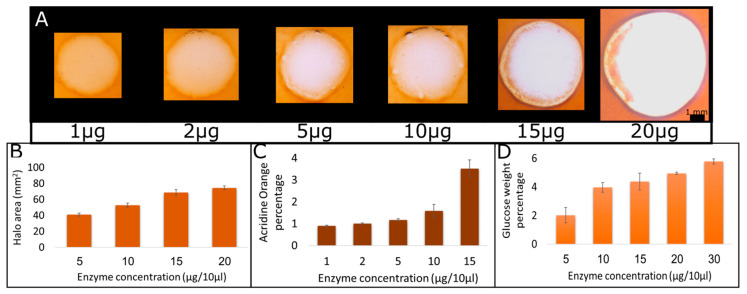
Halo size, dye, and hydrolysis products correlation with enzyme concentration. Halo size was visualized (**A**) and measured (**B**) on films subjected to different enzyme concentrations. Film wash water was analyzed using UV-vis spectroscopy (**C**) and Sumner’s method followed by UV-vis spectroscopy (**D**). CMC dyed films were subjected to 10 μL of different amounts of cellulase. The films were incubated at 37 °C for 2 h and washed with 10 mL water. The halo was visualized (**A**), measured (**B**), and wash water was analyzed using UV-vis spectroscopy. The absorbance at 480 nm was used to calculate acridine orange concentration based on a calibration curve, and the percentage of dye released by hydrolysis is shown (**C**). Additionally, CMC films with no dye were subjected to different amounts of cellulase for 16 h at 37 °C. Wash water was subjected to Sumner’s method. Absorbance was measured using UV-vis. Absorbance at the peak was used to calculate reducing sugars concentration based on a calibration curve and their weight percentage from the CMC film is shown (**D**). The assay was repeated 3 times.

**Figure 3 ijms-21-06593-f003:**
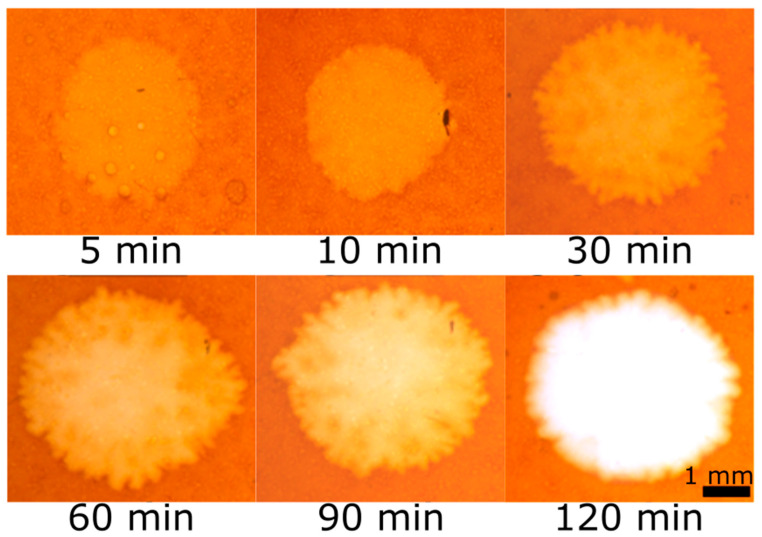
Effect of incubation time. Six different dyed CMC films were subjected to 10 μL containing 10 μg of cellulase. The films were incubated at 37 °C for different time periods, between 5 min and 2 h (as indicated under the tiles); then, they were washed and visualized. The halo becomes bigger and brighter with time. Importantly, a halo can be seen within 5 min, validating this method as a real-time visualization of cellulase activity.

**Figure 4 ijms-21-06593-f004:**
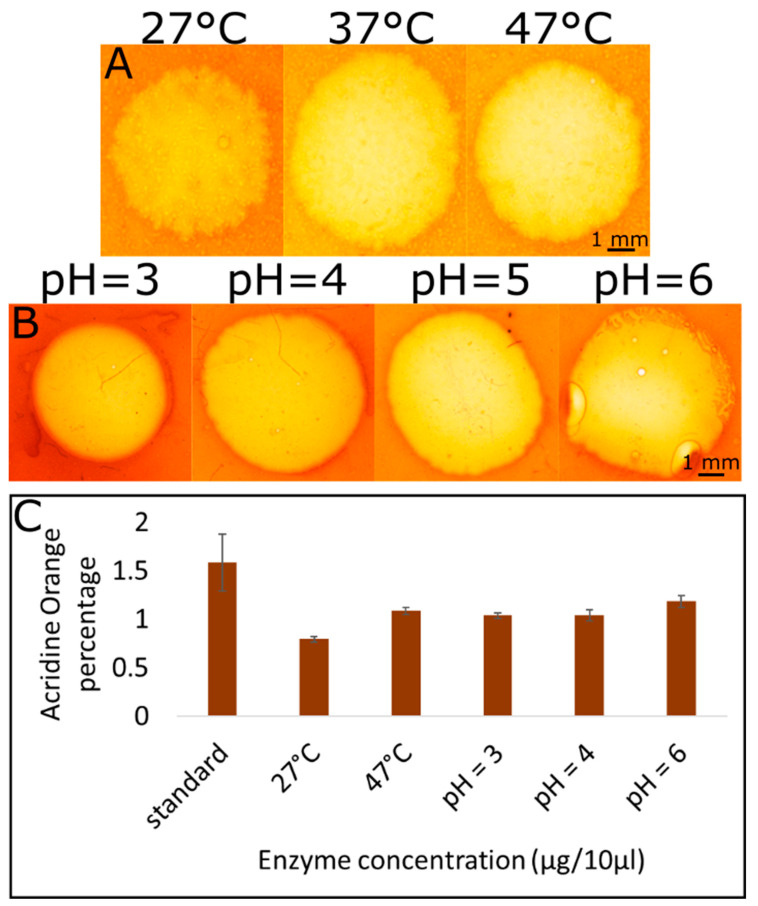
Effect of temperature and pH. (**A**) Dyed CMC films were subjected to 10 μL containing 10 μg of cellulase and incubated for 2 h at 3 different temperatures (27 °C, 37 °C and 47 °C). Then, the films were washed and visualized using a streomicroscope. The halo was biggest and brightest at 37 °C. (**B**) The dyed CMC films were washed with acetate buffer at different pH values (3, 4, 5, and 6) and then subjected to 10 μg cellulase enzyme in 10 μL of the same pH as the acetate buffer. Then, the films were incubated at 37 °C for 2 h followed by washing and visualization. (**C**) UV-vis spectroscopy measuring the percentage of acridine orange released out of the total acridine orange added to the plate at different temperature and pH conditions. Standard conditions denote pH = 5 and a temperature of 37 °C.

**Figure 5 ijms-21-06593-f005:**
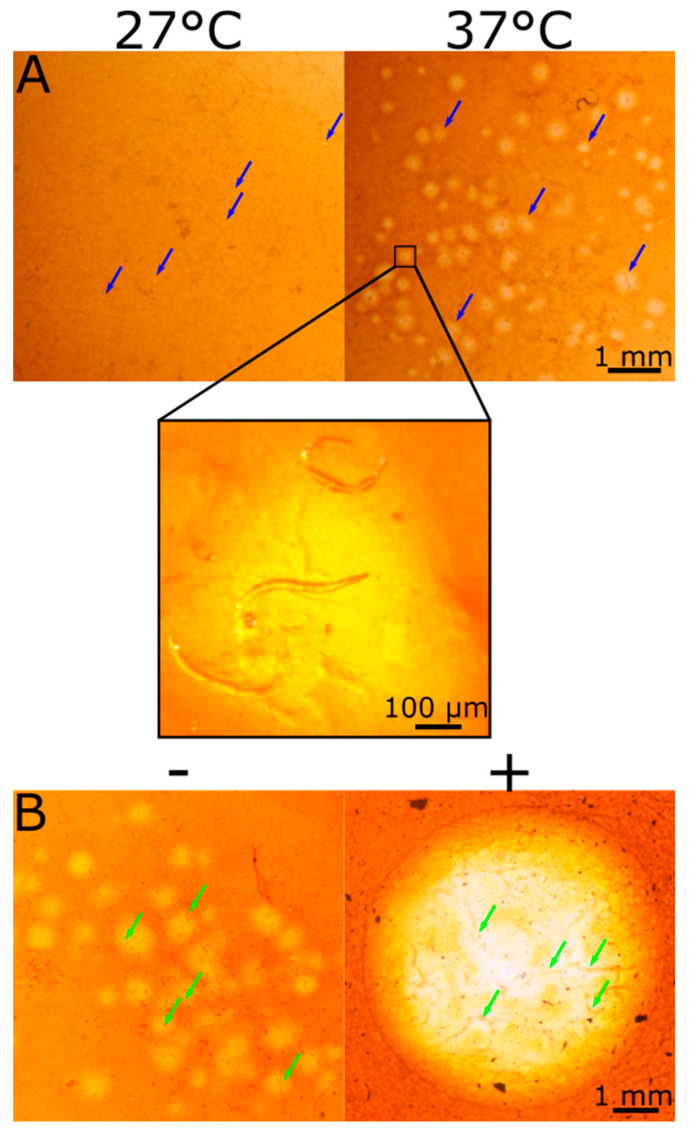
CMC hydrolysis by Root knot nematodes. Approximately 80 nematodes in 50 μL were applied to the center of a dyed CMC film without (**A**) or with (**B**) root extract. (**A**) The films were incubated for 2 h at either 27 °C—the ideal temperature for RKN (left), or 37 °C—the ideal temperature for the CMC hydrolysis reaction (right). The films were washed and visualized. In both temperatures, hydrolysis could be visualized in the form of many spots of color change (marked with blue arrows). The spots on the film incubated at 37 °C were larger and clearer due to the higher efficiency of the hydrolysis reaction in that temperature. An enlargement of one of the spots clearly shows the nematodes located in that spot. (**B**) A comparison between the visualization of cellulase secretion by RKN on a film containing (right side, marked with+) and not containing (left side, marked with−) root extract. The addition of root extract to the film resulted in a clear halo as opposed to the singular spots on the film not containing the root extract, which was probably due to increased cellulase secretion. The nematodes are marked with green arrows.

**Figure 6 ijms-21-06593-f006:**
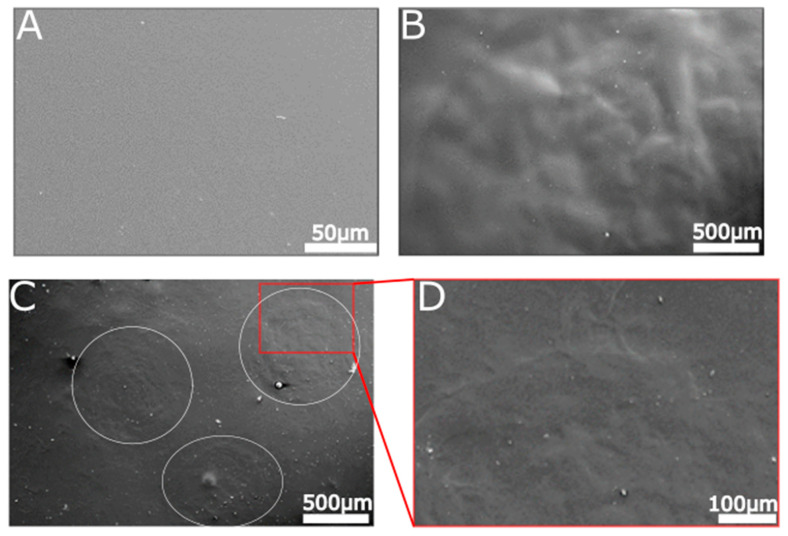
SEM analysis of unhydrolyzed and hydrolyzed CMC films. SEM micrographs of pure CMC film (**A**), CMC film subjected to 10 µg of cellulase enzyme in a volume of 10 μL (**B**), and CMC film subjected to nematodes (**C**,**D**). While the pure CMC film presents a smooth surface, both films subjected to enzyme show a rough pattern. In the film subjected to nematodes, the nematodes location can be seen clearly (**C**, marked with white circles). All films were incubated at 37 °C for 2 h with or without enzyme from any source and washed prior to coating for SEM visualization.

**Figure 7 ijms-21-06593-f007:**
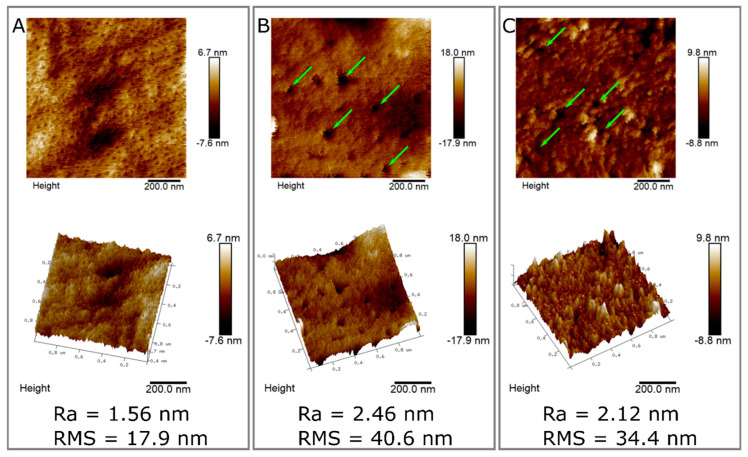
Surface topography measurement of unhydrolyzed and hydrolyzed CMC. Two-dimensional (upper panel) and 3D (lower panel) atomic force microscopy (AFM) images of pure CMC film (**A**), CMC film subjected to 10 µg of cellulase enzyme (**B**), and CMC film subjected to nematodes (**C**). Pure CMC film has small holes and low roughness. Both of the films subjected to enzyme show larger holes (marked with green arrows) and a higher roughness. In the film subjected to nematodes, the holes are smaller, and roughness is lower than the film subjected to cellulase. Roughness values are shown on the bottom. All films were incubated at 37 °C for 2 h with or without enzyme or RKN and washed prior to AFM analysis.

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
