# Peer review of "Real-Time Visualization of Cellulase Activity by Microorganisms on Surface"

_ijms, 2020, doi:10.3390/ijms21186593_

Round 1
Reviewer 1 Report
The purpose of the work is sound and interesting. However, information and redaction is disordered so it is difficult to follow the text. I find difficult to find the results description, the comparison with other authors and conclusions extracted from every result. Here are some additional suggestions:
Line 31: The word “the” is repeated twice
Line 86: Replace “out” by “our”
The language is bad expressed: Please avoid the use of “our”, “we”…Try to use impersonal phrases.
Lines 101-118: There is mixture of introduction, methodology and results. Try to put in order the information.
Figures in general have very low resolution and do not follow the proportionality. Please, include Figures with legends of the same letter size that the text (e.g. Figure 5 letter B is huge).
Materials and method section: I think that materials and method is very extent. I would sun up the section of buffer preparations and standardized protocols, including references when appropriate.
Results:
Have you considered using Calcofluor white stain? There are some reports using this dye.
Author Response
We thank the reviewer for the review.
All lines refer to the lines in the new revised clean version.
Regarding the general comment concerning the disorder, we went over the text and made sure that only little discussion relevant for the understanding of the result in a wider context is added to each results paragraph. Any other comments were moved to the discussion part. For example, the description: “Thus, importantly, our synthetic system is successful in testing the effect of different parameters on microorganism behavior. Our next steps will include separation of root materials and controlled addition of isolated substances. We will also work on adding those substances with a spatial resolution to further link between topographical (physical) and chemical effects on microorganismal behavior on surfaces.” Previously in lines: 264-268, was moved to the discussion in lines: 418-422.
We deleted the first appearance of the word "The" in line 31.
We replaced the word out by our in line 86 (now line 87).
“Lines 101-118: There is mixture of introduction, methodology and results. Try to put in order the information.” - We agree with the reviewer and hence moved what was written in lines 102-110 to an appropriate location in the introduction (now lines 57-57, 93-100).
Regarding figure resolution and proportionality: Most figures were taken from the streomicroscope camera. As we mentioned, we wished to make the setting as simple as possible and hence only used a stereomicroscope. Nonetheless, we went over all figures and found that all proportions are correct, and all letters are on the same size. We suspect there was a problem in the transformation from the .svg files (in which all figures are proportional and in better quality) to the .png files. Upon request, we will happily upload the .svg files and hope this will take care of the problem.
Materials and methods: This reviewer’s concern regarding the materials and methods was the exact opposite of the second reviewers concern. While one thought it was extent the other mentioned more details should be given. We, hence, decided to keep the section as is.
Regarding calcofluor white stain: We did not consider using this dye. This is a dye used to stain cellulose in living organisms and is specific to cellulose. However, we actually preferred a general dye that does not interact with cellulose (or CMC) but can rather be absorbed in the polymeric film and hence released upon film degradation.
Reviewer 2 Report
The manuscript submitted by Kumari et al. entitled “Real Real-time visualization of cellulase activity by microorganisms on surface” presents many interesting data dealing mostly with the development of a new method to detect a real time cellulase activity secreted by microorganisms in interaction with the surface of plant roots. However, there are many points that should be investigated and/or revised:
- The Trichoderma reesei ATCC 26921 purchased from Sigma Aldrich cellulase is used in this study. There is no information about the purification percentage? If author have used a non-purified enzyme, it is difficult to say that the activities observed are related only to cellulase enzyme. The information about the degree of purity should be added in Material and Methods section. Authors should add the SDS-PAGE of the enzyme as a supplementary data.
- Authors should also add the specific activity of the enzyme (IU/mg) at optimal conditions using UV-vis spectroscopy.
- Data presented in Fig. 3 show that the area of the halo obtained after 10 minutes is smaller than 5 minutes. Does authors have an explanation.
- To study the effect of temperature and pH on enzyme activity (Fig. 4), authors have used a qualitative method based on dyed CMC films. Authors should compare the results using UV-spectroscopy. Data should be added in fig. 4.
Minor points:
- Page 14: line 450: After complete dispersion, the CMC solution was added to the dye solution. Is it v/v?
- Page 14: line 464: (1ug/1ul). Please write the concentration and the number of international unit added.
- Page 15: Benedect’s test: please give more information about the test and the principle of the method and why we can observe color change following the addition of enzyme.
- Page 15- Sumner’s method: dH2O, please write deionized water, Line 488: please specify what is the low temperature. Line 490 10 g of 3,5 dinitrosalicilic acid
- Page 15, line 505, RT room temperature
Please avoid if possible the use of abbreviations and give more details about the methods used in the manuscript or add references. It is very interesting to readers to have these details.
Author Response
We thank the reviewer for this positive and thorough review.
All lines refer to the lines in the new revised clean version.
Detailed comments:
- Regarding the purity of the enzyme – according to sigma's website, the enzyme is a mixture containing high cellulase activity with some hemicellulase activity. Clearly, this is not a pure enzyme and it is judged by the activity rather than the purity. This fits to our assay since our assay is intended to use microorganisms which also do not secrete purified enzymes. We hence think it is unnecessary to test for the purity of the enzyme.
- However, we do agree with the reviewer regarding the specific activity of the enzyme, as this is the relevant factor in our case. We, hence, added this data as reported by sigma in the materials and methods section (lines 462-465). This information fits our results as presented in figure 2D and so we added a remark regarding that in the figure description (this fits the reviewer's request to add the specific activity of enzyme at optimal conditions using UV-spectroscopy).
- Regarding Fig. 3, it does seem that the halo after 10 min is slightly smaller than the one after 5 min, but we believe this is just a visual illusion. Our measurements show that both halos are roughly the same size.
- We thank the reviewer for the valuable point regarding the quantification of the reaction under different pH and temperature conditions. The data was added to figure 4 (Figure 4C) and a description was added to the text (lines 222-224) and the figure legend.
Minor points:
- Line 450 (now lines 451-452): The whole CMC solution was added to the whole dye solution. We emphasized this point in the text.
- Line 464: We added the number of units added (line 465).
- Benedict's test was described in more details (lines 489-492).
- Sumner’s method – Line 487 (now line 494): dH2O was changed to deionized water. Line 488 (now line 495): the low temperature was specified. Line 490 (now line 497): the word of was added after 10 g.
- Line 505 (now line 512): RT was changes to room temperature.
Round 2
Reviewer 2 Report
Minor point:
Authors should give information about standard panel in Fig. 4 (Please write pH 5 and 37°C.
Authors have taken into consideration all comments. the manuscript can now be published in international journal of molecular sciences
Author Response
The sentence: :"Standard conditions denote pH=5 and temperature of 37ºC" was added to Figure 4 legend.